# Insertions and deletions as phylogenetic signal in an alignment-free context

Niklas Birth[1], Thomas Dencker[1], Burkhard Morgenstern[1,2,3]*

**1** Department of Bioinformatics, Institute of Microbiology and Genetics, Universisät Göttingen, Göttingen, Germany, **2** Göttingen Center of Molecular Biosciences (GZMB), Göttingen, Germany, **3** Campus-Institute Data Science (CIDAS), Göttingen, Germany

* bmorgen@gwdg.de

**Data Availability Statement:** The source code of our approach is freely available at https://github.com/njbirth/gap-spam. All other relevant data are within the manuscript.

## Abstract

Most methods for phylogenetic tree reconstruction are based on sequence alignments; they infer phylogenies from substitutions that may have occurred at the aligned sequence positions. Gaps in alignments are usually not employed as phylogenetic signal. In this paper, we explore an alignment-free approach that uses insertions and deletions (indels) as an additional source of information for phylogeny inference. For a set of four or more input sequences, we generate so-called *quartet blocks* of four putative homologous segments each. For *pairs* of such quartet blocks involving the same four sequences, we compare the distances between the two blocks in these sequences, to obtain hints about indels that may have happened between the blocks since the respective four sequences have evolved from their last common ancestor. A prototype implementation that we call *Gap-SpaM* is presented to infer phylogenetic trees from these data, using a *quartet-tree* approach or, alternatively, under the *maximum-parsimony* paradigm. This approach should not be regarded as an alternative to established methods, but rather as a complementary source of phylogenetic information. Interestingly, however, our software is able to produce phylogenetic trees from putative indels alone that are comparable to trees obtained with existing alignment-free methods.

## Author summary

Phylogenetic tree inference based on DNA or protein sequence comparison is a fundamental task in computational biology. Given a multiple alignment of a set of input sequences, most approaches compare aligned sequence positions to each other, to find a suitable tree, based on a model of molecular evolution. Insertions and deletions that may have happened since the input sequences evolved from their last common ancestor are ignored by most phylogeny methods. Herein, we show that insertions and deletions can provide an additional source of information for phylogeny inference, and that such information can be obtained with a simple alignment-free approach. We provide an implementation of this idea that we call Gap-SpaM. The proposed approach is complementary to existing phylogeny methods since it is based on a completely different source of

**Funding:** Part of this work was supported by a grant to BM from the VW Foundation [VWZN3157]. The salary for the second author (TD) was partially funded by this grant. The funders had no role in study design, data collection and analysis, decision to publish, or preparation of the manuscript.

**Competing interests:** The authors have declared that no competing interests exist.

information. It is, thus, not meant to be an alternative to those existing methods but rather as a possible additional source of information for tree inference.

This is a *PLOS Computational Biology* Methods paper.

## 1 Introduction

Most phylogenetic studies are based on multiple sequence alignments (MSAs), either of partial or complete genomes or of individual genes or proteins. If MSAs of multiple genes or proteins are used, there are two possibilities to infer a phylogenetic tree: (1) the alignments can be concatenated to form a so-called *superalignment* or *supermatrix*. Tree building methods such as *Maximum-Likelihood* [1, 2], *Bayesian Approaches* [3] or *Maximum-Parsimony* [4–6] can then be applied to these superalignments. (2) One can calculate a separate tree for each gene or protein family and then use a *supertree approach* [7] to amalgamate these different trees into one final tree, with methods such as *ASTRAL* [8] or *MRP* [9].

Multiple sequence alignments usually contain gaps representing insertions or deletions (*indels*) that are assumed to have happened since the aligned sequences evolved from their last common ancestor. Gaps, however, are usually not used for phylogeny reconstruction. Most of the above tree-reconstruction methods are based on substitution models for nucleotide or amino-acid residues. Here, alignment columns with gaps are either completely ignored, or gaps are treated as 'missing information', for example in the frequently used tool *PAUP\** [6]. Some models have been proposed that can include gaps in a *Maximum-Likelihood* setting, such as *TKF91* [10] and TKF92 [11], see also [12–14]. Unfortunately, these models do not scale well to genomic data. Thus, indels are rarely used as a source of information for the phylogenetic analysis.

In those studies that actually make use of indels, this additional information is usually encoded in some simple manner. The most straightforward way of doing this is to treat the gap character as a fifth character for DNA comparison, or as a 21st character in protein comparison, respectively. This means that the lengths of gaps are not explicitly considered, so a gap of length $\ell > 1$ is considered to represent $\ell$ independent insertion or deletion events. Some more issues with this approach are discussed in [15]; these authors introduced the 'simple encoding' of indel data as an alternative. For every indel in the multiple sequence alignment, an additional column is appended. This column contains a present/absent encoding for an indel event which is defined as a gap with given start and end positions. If a longer gap is fully contained in a shorter gap in another sequence, it is considered as *missing information*. Such a simple binary encoding is an effective way of using the length of the indels to gain additional information and can be used in some *maximum-parsimony* framework. A disadvantage of these approaches is their relatively long runtime. The above authors also proposed a more complex encoding of gaps [15] which they further refined in a subsequent paper [16]. The commonly used approaches to encode gaps for phylogeny reconstruction are compared in [17].

The 'simple encoding' of gaps has been used in many studies; one recent study obtained additional information on the phylogeny of Neoaves which was hypothesized to have a 'hard polytomy' [18]. Despite such successes, indel information is still largely ignored in phylogeny

reconstruction. Oftentimes, it is unclear whether using indels is worth the large overhead and increased runtime. On the hand, it has also been shown that gaps can contain substantial phylogenetic information [19].

All of the above mentioned approaches to use indel information for phylogeny reconstruction require MSAs of the compared sequences. Nowadays, the amount of the available molecular data is rapidly increasing, due to the progress in next-generation sequencing technologies. If the size of the analyzed sequences increases, calculating multiple sequence alignments quickly becomes too time consuming. Thus, in order to provide faster and more convenient methods to phylogenetic reconstruction, many alignment-free approaches have been proposed in recent years. Most of these approaches calculate pairwise distances between sequences, based on sequence features such as *k*-mer frequencies [20–22] or the number [23] or length [24–26] of word matches. Distance methods such as *Neighbor-Joining* [27] or *BIONJ* [28] can then reconstruct phylogenetic trees from the calculated distances. For an overview, the reader is referred to recent reviews of alignment-free methods [29–31].

Some recently proposed alignment-free methods use inexact word matches between pairs of sequences [32–34], where mismatches are allowed to some degree. Such word matches can be considered as pairwise, gap-free 'mini-alignments'. So, strictly spoken, these methods are not 'alignment-free'. In the literature, they are still called 'alignment-free', as they circumvent the need to calculate full sequence alignments of the compared sequences. The advantage of such 'mini-alignments' is that inexact word matches can be found almost as efficiently as exact word matches, by adapting standard word-matching algorithms.

A number of these methods use so-called *spaced-words* [22, 35, 36]. A spaced-word is a word that, in addition to nucleotide or amino-acid symbols, contains *wildcard* characters at certain positions that are specified by a pre-defined binary pattern *P* representing 'match positions' and 'don't-care positions', see Fig 1 for an example. If the same 'spaced word' occurs in two different sequences, this is called a *Spaced-word Match* or *SpaM*, for short. One way of using spaced-word matches–or other types of inexact word matches–in alignment-free sequence comparison is to use them as a proxy for full alignments, to estimate the number of mismatches per position in the (unknown) full sequence alignment. This idea has been implemented in the software *Filtered Spaced Word Matches (FSWM)* [34]; it has also been applied to protein sequences [37], and to unassembled reads [38].

In such approaches, it is crucial to use only those *SpaMs* that align *homologous* segments of the compared sequences and to discard random *SpaMs*. *FSWM* and related programs *filter out* non-homologous *SpaMs* by comparing the residues aligned to each other at the *don't-care positions* of the *SpaMs*. As shown in Fig 1, a score can be calculated based on these residue pairs, and all *SpaMs* with a score below a certain threshold are discarded. As we have shown in previous papers, this approach can reliably distinguish between homologous and background *SpaMs* [34]. Other approaches have been proposed recently, that use the *number* of *SpaMs* to estimate phylogenetic distances between DNA sequences [36, 39], see [40] for a review of the various *SpaM*-based methods.

*Multi-SpaM* [41] is a recent extension of the *SpaM* approach to *multiple* sequence comparison. For a set of four or more input sequences, and for a binary pattern *P*, *Multi-SpaM* finds occurrences of the same spaced word with respect to *P* in *four* different input sequences. Such a spaced-word match is called a *quartet P-block*, or *quartet block*, for short. A *quartet block*, thus, consists of four occurrences of the same spaced-word, with respect to a specific pattern *P*, as in Fig 2. For each such block, *Multi-SpaM* identifies an optimal quartet tree topology based on the nucleotides aligned to each other at the *don't-care* position of *P*, using the program *RAxML* [1]. Finally, the quartet trees calculated in this way are used to find a supertree of the full set of input sequences. To this end, *Multi-SpaM* uses the program *Quartet MaxCut* [42].

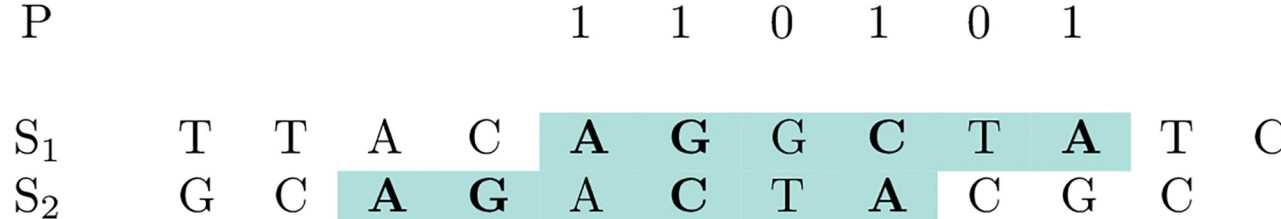

**Fig 1. Binary pattern *P* = '110101" ('1') and *don't-care positions* ('0') and a *spaced word* 'A G * C * A' with respect to *P*, occurring in sequences $S_1$ and $S_2$.** The occurrence of the same spaced word in two different sequences is called a *Spaced-word Match (SpaM)*. A *SpaM* w.r.t. *P* is, thus, a local gap-free alignment where matching residues are aligned at the *match positions* of *P*, while mismatches are possible at the *don't-care positions*. In the above toy example, we find at the don't care positions one mismatch (A-G) and one match (T-T). A *score* can be calculated for each *SpaM* based on the residues aligned to each other at the *don't-care positions*. If the number of *don't-care positions* in the underlying pattern *P* is sufficiently large, 'homologous' *SpaMs* can be reliably distinguished from background by their scores [34].

In the present paper, we use *pairs* of *quartet blocks* involving the same four sequences. We consider the distances between two blocks in the four sequences, to obtain hints about potential insertions and deletions that may have occurred between two quartet blocks. If these distances are different for two of the sequences, this would indicate that an insertion or deletion has happened since these sequences evolved from their last common ancestor. The distances between two quartet blocks can therefore support one of three possible quartet topologies for the four involved sequences. If, for example, in a pair of quartet blocks involving sequences $S_i$, $S_j$, $S_k$, $S_l$, the distance between these blocks is equal in $S_i$ and $S_j$ as well as in $S_k$ and $S_l$ but the distance in $S_i$ and $S_j$ is different from the one in $S_k$ and $S_l$, this would support a quartet tree where $S_i$ and $S_j$ are neighbours, as well as $S_k$ and $S_l$; an example is shown in Fig 3.

To evaluate the phylogenetic signal that is contained in such pairs of quartet blocks, we first evaluate the inferred quartet topologies directly, by comparing them to trusted reference trees. Next, we use two different methods to infer a phylogenetic tree for the full set of input sequences, based on the distances between quartet blocks. (*A*) We calculate super trees based on the inferred quartet trees using the software *Quartet MaxCut*. (*B*) We use distances between pairs of blocks as characters in a *maximum-parsimony* setting, to find a tree that minimizes the number of insertions and deletions that have to be assumed, given the different distances between the quartet blocks. We evaluate these approaches on data sets that are commonly used as benchmark data in alignment-free sequence comparison. Our evaluation shows that the

| Sequence | | | | | | | | | | | | |
|---|---|---|---|---|---|---|---|---|---|---|---|---|
| $S_1$ | T | A | G | A | T | G | C | C | A | T | A | A | T |
| $S_2$ | T | T | A | C | **A** | **G** | G | **C** | A | **A** | T | C | |
| $S_3$ | G | C | **A** | **G** | A | **C** | G | **A** | C | G | C | | |
| $S_4$ | T | A | G | A | C | A | A | G | T | C | C | T | |
| $S_5$ | A | T | T | **A** | **G** | G | **C** | C | **A** | C | T | C | A |
| $S_6$ | A | A | G | G | C | A | A | C | T | C | G | | |
| $S_7$ | A | **A** | **G** | T | **C** | A | **A** | C | T | C | G | T | |
| $S_8$ | G | C | T | A | G | C | T | T | C | A | T | C | A |
| $S_9$ | C | T | T | A | A | A | C | G | G | C | T | T | |

**Fig 2. Quartet block with respect to the binary pattern 110101 representing *match positions* ('1') and *don't-care positions* ('0').** The shown *quartet block* involves sequences $S_2$, $S_3$, $S_5$, $S_7$; the spaced word '**A G** * **C** * **A**' occurs in all four sequences. A *quartet block* can be seen as a local, gap-free four-way alignment with matching residues at the *match positions* and possible mismatches at the *don't-care positions* of the underlying binary pattern. Note that this is a toy example, in practice we are using binary patterns of length 110 with 10 *match* and 100 *don't-care* positions.

| Sequence | | | | | | | | | | | | | | | | | Distance $D_i$ |
|---|---|---|---|---|---|---|---|---|---|---|---|---|---|---|---|---|---|
| $S_1$ | A | A | G | A | C | G | T | T | A | C | A | C | G | A | T | | |
| $S_2$ | T | C | **A** | **G** | G | **C** | A | A | **C** | **G** | **G** | **T** | **A** | C | | | $D_2 = 2$ |
| $S_3$ | T | T | G | A | G | A | C | A | T | C | C | G | A | T | C | A | |
| $S_4$ | C | **A** | **G** | **A** | **C** | A | C | T | **C** | **G** | **G** | **T** | **A** | T | A | | $D_4 = 3$ |
| $S_5$ | A | A | T | A | **A** | **G** | T | **C** | A | T | **C** | **A** | **G** | **T** | **A** | | $D_5 = 2$ |
| $S_6$ | G | A | C | T | C | G | T | T | C | C | C | G | A | C | A | | |
| $S_7$ | G | T | G | C | C | A | A | C | C | C | A | G | C | C | C | | |
| $S_8$ | C | G | **A** | **G** | **T** | **C** | A | A | T | **C** | **A** | **G** | **T** | **A** | C | T | $D_8 = 3$ |
| $S_9$ | A | C | C | C | T | C | C | C | G | A | G | C | A | C | A | A | |

**Fig 3. Two quartet blocks $B_1$ and $B_2$ (in green and purple) with respect to binary patterns 1101 and 10111, and with the matching spaced words 'A G * C' and 'C * G T A', respectively, involving the same four sequences $S_2$, $S_4$, $S_5$, $S_8$.** The distances between $B_1$ and $B_2$ in these sequences are $D_2 = D_5 = 2$ and $D_4 = D_8 = 3$. In the sense of *maximum parsimony*, these distances would support the quartet topology $S_2S_5|S_4S_8$, since this topology would require only one insertion/deletion (indel) event to explain the distances $D_i$ while the alternative two quartet topologies for the involved sequences would require two indel events. With our terminology, we say that this topology is *strongly* supported by the four distance values $D_i$.

majority of the inferred quartet trees is correct and should therefore be useful additional information for phylogeny reconstruction. Moreover, the quality of the trees that we can infer from our quartet block pairs alone is roughly comparable to the quality of trees obtained with existing alignment-free methods.

The goal of our study is to show that insertions and deletions can be used as phylogenetic signal in an alignment-free context. Note that the information from putative indels is *complementary* to the information used in standard phylogeny approaches where aligned residues are used to infer substitutions that may have happened in the evolution of the sequences. Consequently, our approach is not competing with these existing methods but may be used as *additional* evidence that might support or call into question phylogenies inferred by more traditional approaches.

## 2 Methods

### 2.1 Spaced words, quartet blocks and distances between quartet blocks

We are using standard notation from stringology as defined, for example, in [43]. For a sequence $S$ over some alphabet, $S(i)$ denotes the $i$-th symbol of $S$. In order to investigate the information that can be obtained from putative indels in an alignment-free context, we use the *P-blocks* generated by the program *Multi-SpaM* [41]. At the start of every run, a binary pattern $P \in \{0, 1\}^\ell$ is specified for some integer $\ell$. Here, a "1" in $P$ denotes a *match position*, a "0" stands for a *don't-care position*. The number of *match positions* in $P$ is called its *weight* and is denoted by $w$. By default, we are using parameter values $\ell = 110$ and $w = 10$, so by default the pattern $P$ has 100 *don't-care* positions.

A *spaced word* $W$ with respect to a pattern $P$ is a word over the alphabet $\{A, C, G, T\} \cup \{*\}$ of the same length as $P$, and with $W(i) = *$ if and only if $i$ is a *don't care position* of $P$, i.e. if $P(i) = 0$. If $S$ is a sequence of length $N$ over the nucleotide alphabet $\{A, C, G, T\}$, and $W$ is a spaced word, we say that $W$ *occurs* at some position $i \in \{1, \ldots, \ell\}$, if $S(i + j - 1) = W(j)$ for every match position $j$ in $P$. For two sequences $S$ and $S'$ and positions $i$ and $i'$ in $S$ and $S'$, respectively, we say that there is a *spaced-word match (SpaM)* between $S$ and $S'$ at $(i, i')$, if the same spaced word $W$ occurs at $i$ in $S$ and at $i'$ in $S'$. A SpaM can be considered as a local pairwise alignment without gaps. Given a nucleotide substitution matrix, the *score* of a spaced-word match is defined as the sum of the substitution scores of the nucleotides aligned to each other at the

*don't-care* positions of the underlying pattern *P*. In *FSWM* and *Multi-SpaM*, we are using a substitution matrix described in [44]. In *FSWM*, only *SpaMs* with positive scores are used. It has been shown that this *SpaM-filtering* step can effectively eliminate most random spaced-word matches [34].

For a set of $\geq 4$ input sequences and a binary pattern *P* of length $\ell$, the program *Multi-SpaM* is based on *quartet (P)-blocks*, where a quartet block is defined as four occurrences of some spaced word *W* in four different sequences, see Fig 2 for an example. A quartet block *B* can, thus, be considered as a local gap-free four-way alignment, aligning length-$\ell$ segments of four sequences; we say that *B* 'involves' these four sequences. To exclude spurious random quartet blocks, *Multi-SpaM* removes quartet blocks with a low degree of similarity between the aligned segments. Technically, a quartet block is required to contain one occurrence of the spaced-word *W*, such that the other three occurrences of *W* have positive similarity scores with this first occurrence. For a given nucleotide substitution matrix, the similarity score of two spaced words (with respect to the same pattern *P*) is defined as the sum of the substitution scores of the nucleotides aligned to each other at the *don't-care* positions of *P*.

## 2.2 Phylogeny inference using distances between quartet blocks

In this paper, we are considering *pairs* of quartet blocks involving the same four sequences, and we are using the distances between the two blocks in these sequences as phylogenetic signal. The first block in a block pair is called the *reference block*. To find reference blocks, we use the program *Multi-SpaM*. This program identifies quartet blocks with respect to a binary pattern $P_1$, as explained in the first section of this paper. Here, a *score* is calculated for each quartet block, based on the don't-care positions, to exclude random spaced-word matches, as detailed above. For each *reference block* with a score above the threshold, our new approach then searches for a second quartet block, involving the same four sequences, possibly with a different pattern $P_2$, and within a window of *L* nucleotides in each sequence, to the right of the reference block. By default, we are using a window size of *L* = 500. For the second block, we do not calculate a score, since the probability of finding a quartet block within such a window by chance is very small.

Let us consider two *quartet blocks*—a reference block $B_1$ and a corresponding second block $B_2$ as described above –, with respect to patterns $P_1$ and $P_2$, respectively, involving the same four sequences $S_i, S_j, S_k, S_l$. By definition, $B_1$ is strictly to the left of $B_2$, in the sense that the last position of $B_1$ is smaller than the first position of $B_2$ in all four sequences. Next, let $D_t$ be the distance between $B_1$ and $B_2$ in sequence $S_t$, $t = i, \ldots, l$. More formally, if in sequence $S_t$ block $B_1$ starts at position $k_1$ and block $B_2$ starts at position $k_2$, then we define $D_t$ to be $k_2 - k_1 - \ell_1$, where $\ell_1$ is the length of the pattern $P_1$. In other words, $D_t$ is the length of the segment between $B_1$ and $B_2$ in $S_t$, see Figs 3 and 4 for examples. As explained, we can assume that the blocks $B_1$ and $B_2$ are representing true homologies, i.e. for each of them the respective segments go back to a common ancestor in evolution. Then, if we find for two sequences, say $S_i$ and $S_j$, that their distances $D_i$ and $D_j$ between $B_1$ and $B_2$ are different from each other, this would imply that at least one insertion or deletion must have happened since $S_i$ and $S_j$ have evolved from their last common ancestor. If, by contrast, the $D_i = D_j$ holds, no such insertion or deletion needs to be assumed.

There are three possible fully resolved (i.e. binary) quartet topologies for the four sequences $S_i, \ldots, S_l$ that we denote by $S_iS_j|S_kS_l$ etc. In the sense of the *parsimony* paradigm, we can consider the distance between two blocks as a *character* and $D_t$ as the corresponding *character state* associated with sequence $S_t$. If two distances, say $D_i$ and $D_j$, are equal, and the other two distances, $D_k$ and $D_l$ are also equal to each other, but different from $S_i$ and $S_j$, respectively, this

| Sequence | | | | | | | | | | | | | | | Distance $D_i$ |
|---|---|---|---|---|---|---|---|---|---|---|---|---|---|---|---|
| $S_1$ | G | **A** | **G** | G | **C** | A | A | **C** | G | **G** | **T** | **A** | C | T | T | $D_1 = 2$ |
| $S_2$ | G | G | A | C | A | C | G | T | T | A | C | C | G | A | | |
| $S_3$ | T | T | G | A | G | A | C | A | T | C | C | G | A | T | C | |
| $S_4$ | A | A | T | A | **A** | **G** | **T** | **C** | A | T | **C** | **A** | **G** | **T** | **A** | $D_4 = 2$ |
| $S_5$ | C | **A** | **G** | A | **C** | A | A | C | T | **C** | **G** | **G** | **T** | **A** | | $D_5 = 4$ |
| $S_6$ | C | G | **A** | **G** | T | **C** | A | A | T | **C** | **A** | **G** | **T** | **A** | C | $D_6 = 3$ |
| $S_7$ | G | A | C | T | C | G | T | T | C | C | C | G | A | C | A | |
| $S_8$ | C | T | C | G | T | T | C | C | C | G | A | C | A | A | | |

**Fig 4. Two quartet blocks, similar as in Fig 3, but involving $S_1$, $S_4$, $S_5$, $S_6$, and with distances $D_1 = D_4 = 2$, $D_5 = 3$ and $D_6 = 4$.** Here, we say that the distances *weakly* support the topology $S_1S_4|S_5S_6$, since only $D_1$ and $D_4$ are equal, while $D_5$ and $D_6$ are different from each other and from $D_1$ and $D_4$.

would support the tree topology $S_iS_j|S_kS_l$: with this topology, one would have to assume only one insertion or deletion to explain the character states, while for $S_iS_k|S_jS_l$ or $S_iS_l|S_jS_k$, two insertions or deletions would have to be assumed. In this situation—i.e. if we have $D_i = D_j \neq D_k = D_l$ –, we say that the pair $(B_1, B_2)$ *strongly* supports topology $S_iS_j|S_kS_l$.

Next, we consider the situation where two of the distances are equal, say $D_i = D_j$, and $D_k$ and $D_l$ would be different from each other, and also different from $D_i$ and $D_j$. From a parsimony point-of-view, all three topologies would be equally good in this case, since each of them would require two insertions or deletions. It may still seem more plausible, however, to prefer the topology $S_iS_j|S_kS_l$ over the two alternative topologies. In fact, if we would use a simple probabilistic model where an insertion/deletion event has a fixed probability $p$, with $0 < p < 0.5$, along each branch of the topology, then it is easy to see that the topology $S_iS_j|S_kS_l$ would have a higher likelihood than the two alternative topologies. In this situation, we say that the pair $(B_1, B_2)$ *weakly* supports the topology $S_iS_j|S_kS_l$. Finally, we call a pair of quartet blocks *informative*, if it–strongly or weakly—supports one of the three quartet topologies for the involved four sequences.

For a set of input sequences $S_1, \ldots, S_N$, $N \geq 4$, we implemented two different ways of inferring phylogenetic trees from quartet-block pairs. With the first method, we calculate the *quartet topology* for each quartet-block pair that supports one of the three possible quartet topologies. We then calculate a *supertree* from these topologies. Here, we use the program *Quartet MaxCut* [42, 45] that we already used in our previous software *Multi-SpaM* where we inferred quartet topologies from the nucleotides aligned at the *don't-care* positions of quartet blocks.

Our second method uses the distances between quartet blocks as input for *Maximum-Parsimony* [4, 5]. To this end, we generate a character matrix as follows: the rows of the matrix correspond, as usual, to the input sequences, and each informative quartet block pair corresponds to one column. The distances between the two quartet blocks are encoded by characters '0', '1' and '2', such that equal distances in an informative quartet-block pair are encoded by the same character (this encoding is necessary, since some parsimony programs accept only simple characters as input, so we cannot use the distances themselves as characters in the matrix). For sequences not involved in a quartet-block pair, the corresponding entry in the matrix is empty and is considered as 'missing information'. In Fig 3, for example, the entries for $S_2$, $S_4$, $S_5$, $S_8$ would be '0', '1', '0', '1', respectively; in Fig 4, the entries for $S_1$, $S_4$, $S_5$, $S_6$ would be '0', '0', '1', '2'.

Fig 5 shows an informative block pair, a character matrix encoding the distances $D_i$ for this block pair in the first column and the distances for three additional hypothetical block pairs in columns 2 to 4, together with a tree topology inferred from this matrix with *maximum*

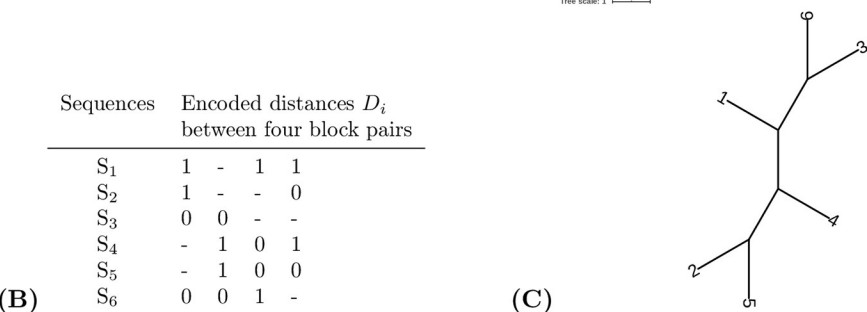

| Sequences | | | | | | | | | | | | | Distances $D_i$ |
|---|---|---|---|---|---|---|---|---|---|---|---|---|---|
| $S_1$ | T | C | A | G | G | C | A | C | G | G | T | A | C | $D_1 = 2$ |
| $S_2$ | A | T | A | A | G | T | A | T | C | A | G | T | | $D_2 = 2$ |
| $S_3$ | C | A | G | A | C | A | T | C | G | G | T | A | T | $D_3 = 3$ |
| $S_4$ | T | T | G | A | G | A | C | A | T | C | G | A | T | |
| $S_5$ | G | A | C | T | C | T | T | C | C | C | G | A | | |
| $S_6$ | C | A | G | A | G | T | A | A | T | C | A | G | T | $D_6 = 3$ |

**(A)**

| Sequences | Encoded distances $D_i$ between four block pairs | | | |
|---|---|---|---|---|
| $S_1$ | 1 | - | 1 | 1 |
| $S_2$ | 1 | - | - | 0 |
| $S_3$ | 0 | 0 | - | - |
| $S_4$ | - | 1 | 0 | 1 |
| $S_5$ | - | 1 | 0 | 0 |
| $S_6$ | 0 | 0 | 1 | - |

**(B)**

**(C)**

**Fig 5.** **(A)** Single block pair in a set of 6 sequences and distances $D_i$, **(B)** character matrix encoding distances $D_i$ from four different quartet-block pairs and **(C)** tree topology, calculated from this matrix with *maximum parsimony*. Each column in the matrix represents one informative block pair. For the four sequences involved in a block pair, the distances $D_i$ are represented by characters '0' and '1', such that equal distances are represented by the same character. The characters themselves are arbitrary, the matrix only encodes if the distances $D_i$ between two blocks are equal or different in the four involved sequences. Dashes in a column represent 'missing information', for sequences that are not involved in the respective quartet-block pair. The quartet-block pair in **(A)** would be represented by the first column of the matrix **(B)**, as we have $D_1 = D_2 \neq D_3 = D_6$. Thus, for $S_1$ and $S_2$ we have the same (arbitrary) symbol '1', while $S_3$ and $S_6$ we have the symbol '0'. Since $S_4$ and $S_5$ are not involved in this quartet-block pair, they have dashes in the first column, representing 'missing information'. The matrix represents four quartet-block pairs that *strongly* support one quartet topology, namely column 1 supporting $S_1 S_2 | S_3 S_6$, column 2 supporting $S_3 S_6 | S_4 S_5$, column 3 supporting $S_1 S_6 | S_4 S_5$ and column 4 supporting $S_1 S_4 | S_2 S_5$.

*parsimony*. Here, we used the the program *pars* form the *PHYLIP* package [46]. Note that all four block pairs in the matrix *strongly* support one of the three possible quartet topologies, since a block pair that only weakly supports a topology would not be informative in the sense of the parsimony principle. Therefore, in each of the four block pairs, we have only two different distances, and we need only two characters, '0' and '1'.

In order to find suitable quartet-block pairs for the two described approaches, we are using our software *Multi-SpaM*. This program samples up to 1 million quartet blocks. We use the quartet blocks generated by *Multi-SpaM* as *reference blocks*, and for each reference block $B_1$, we search for a second block in a window of $L$ nucleotides to the right of $B_1$ for a second block $B_2$ involving the same four sequences (default: $L = 500$). We use the first block that we find in this window, provided that the involved spaced-word matches are *unique* within the window. If the pair $(B_1, B_2)$ supports a topology of the involved four sequences—either strongly or weakly –, we use this block pair, otherwise the pair $(B_1, B_2)$ is discarded.

## 3 Test results

In order to evaluate the above described approaches to phylogeny reconstruction, we used five sets of genome sequences from *AF-Project* [47] that are frequently used as benchmark data for alignment-free methods. In addition, we used a set of *Wolbachia* genomes [48], and sets of

**Table 1. Benchmark data sets used to evaluate our approach with number of sequences and average sequence length.** The last column contains the average phylogenetic distance in the respective data set, measured as substitutions per position, estimated by the program *FSWM*.

|  | # seq | avg. length | avg. distance |
|---|---|---|---|
| *E. coli* 29 | 29 | 4,895,247 bp | 0.02 |
| *E. coli* 27 | 27 | 4,905,896 bp | 0.02 |
| Fish mito | 25 | 16,623 bp | 0.17 |
| *Wolbachia* | 19 | 1,321,494 bp | 0.06 |
| *Yersinia* | 8 | 4,605,552 bp | 0.002 |
| Plants | 14 | 337,515,688 bp | 0.54 |
| *Piroplasmida* mito | 19 | 6,571 bp | 0.28 |
| *Termites* mito | 21 | 15,908 bp | 0.26 |

mitochondrial genomes from *Piroplasmida* [49] and from *Termites* [50]. These data sets are summarized in Table 1; for each data set, the number of genome sequences and their average length is given, together with the average pairwise phylogenetic distance in the set. As distance measure, we used the number of substitutions per position, estimated with our program *FSWM*. For these sets of genomes, trusted phylogenetic trees are available that can be used as reference trees; these genomes have also been used as benchmark data to evaluate *Multi-SpaM* [41].

Note that our indel-based approach is not meant to be an alternative to existing phylogeny approaches that are based on substitutions. Since we are using a complementary source of information, we are not competing with those existing methods, but we wanted to know if our approach might be useful as an *additional* input for tree reconstruction. The comparison with alternative alignment-free phylogeny methods in this section is not done to find out which approach performs better—we rather wanted to find out if or to what extent our indel-based approach can provide relevant information for phylogeny inference at all.

## 3.1 Quartet trees from quartet-block distances

First, we tested, how many *informative* quartet block pairs we could find, i.e. how many of the identified quartet-block pairs would either *strongly* or *weakly* support one of the three possible quartet topologies for the corresponding four sequences.

As explained above, for each set of genome sequences, we first sampled up to 1,000,000 quartet blocks with *Multi-SpaM* [41], we call these blocks the 'reference blocks'. For each of these blocks, we then searched for a second block in a window of 500 *nt* to the right of the reference block. For the second block, we used a pattern $P = 1111111$, i.e. we generated blocks of exact word matches of length seven. If no second block could be found in the window, the reference block was discarded. Table 2 shows the percentage of informative quartet block pairs, among the quartet block pairs that we used. To evaluate the correctness of the obtained quartet topologies, we compared them to the topologies of the respective quartet sub-trees of the reference trees using the *Robinson-Foulds (RF)* distance [51] between the two quartet topologies. If the *RF* distance is zero, the inferred quartet topology is in accordance with the reference tree. To compare the obtained quartet topologies to the full reference trees, we used Sarah Lutteropp's program *Quartet Check* that is available through GitHub [52]. We slightly modified the original code to adapt it to our purposes; the modified code used in our study is also available through GitHub [53].

We want to use the quartet trees that we obtain from informative quartet block pairs, to generate a tree of the full set of input sequences. Therefore, it is not sufficient for us to have a

**Table 2. Test results on different sets of genomes.** As benchmark data, we used five sets of genome sequences from *AF-Project* [47] and sets of genomes from *Wolbachia* [48], *Piroplasmida* [49] and *Termites* [50]. For each data set, we generated up to 1,000,000 pairs of quartet blocks as described in the main text. The table shows the number of *informative* block pairs ('# inf bp'), i.e. the number of block pairs for which we obtained either strong or weak support for one of the three possible quartet topologies of the involved sequences. In addition we show the percentage of *correct* quartet topologies (with respect to the respective reference tree), out of all informative block pairs, as well as the 'coverage' by quartet blocks, i.e. the percentage of sequence quartets for which we found at least one informative block pair. Standard deviations are shown in parentheses. For each data set, we obtained 1,000,000 block pairs, except for the three sets of mitochondrial genomes, where it was not possible to find this number of block pairs.

| | Strong support | | | Weak support | | | Strong and weak combined | | |
|---|---|---|---|---|---|---|---|---|---|
| | # inf bp | % corr | % cov | # inf bp | % corr | % cov | # inf bp | % corr | % cov |
| *E. coli* 29 | 54,796 (335) | 80.43 (0.14) | 64.35 (0.44) | 15,932 (277) | 56.59 (0.39) | 36.91 (0.61) | 70,727 (478) | 75.06 (0.18) | 72.56 (0.4) |
| *E. coli* 27 | 54,721 (315) | 78.63 (0.28) | 72.54 (0.35) | 17,759 (208) | 54.65 (0.41) | 47.52 (0.29) | 72,480 (486) | 72.75 (0.27) | 79.65 (0.34) |
| Fish mito | 7,701 (411) | 66.66 (1.23) | 27.4 (2.01) | 10,775 (585) | 58.88 (0.67) | 36.7 (2.36) | 18,476 (985) | 62.12 (0.84) | 48.55 (2.84) |
| *Wolbachia* | 96,607 (762) | 92.47 (0.1) | 94.76 (0.34) | 38,007 (467) | 71.76 (0.25) | 87.4 (0.48) | 134,614 (1005) | 86.62 (0.11) | 99.1 (0.14) |
| *Yersinia* | 5,696 (107) | 44.12 (0.68) | 100 (0) | 5,677 (87) | 41.57 (0.54) | 100 (0) | 11,373 (162) | 42.85 (0.4) | 100 (0) |
| Plants | 15,617 (756) | 82.67 (0.47) | 84.05 (4.09) | 99,178 (1291) | 76.7 (0.5) | 99.93 (0.11) | 114,795 (1983) | 77.5 (0.45) | 99.96 (0.07) |
| *Piroplasmida* mito | 519 (56) | 63.78 (1.8) | 4.81 (0.53) | 1,087 (84) | 43.89 (0.65) | 8.89 (0.655) | 1,599 (138) | 50.29 (1.04) | 10.56 (0.87) |
| *Termites* mito | 1,865 (92) | 47.98 (0.93) | 18.07 (0.93) | 3,230 (132) | 43.82 (0.69) | 28.3 (0.92) | 5,095 (214) | 45.34 (0.59) | 37.45 (1.38) |

high percentage of correct quartet trees, but we also want to know how many of the sequence quartets are covered by these quartet trees. Generally, the results of super-tree methods depend on the *coverage* of the used quartet topologies [54, 55]. For a set of $N$ input sequences, there are $\binom{N}{4}$ possible 'sequence quartets', i.e. sets of four sequences. Ideally, for every such set, we should have at least one quartet tree, in order to find the correct super tree. Table 2 reports the *quartet coverage*, i.e. the percentage of all sequence quartets, for which we obtained at least one quartet tree.

Note that *Multi-SpaM* uses randomly sampled quartet blocks, the program can thus return different results for the same set of input sequences. We therefore performed 10 program runs on each set of sequences and report the average *correctness* and *coverage* of these test runs.

## 3.2 Full phylogeny reconstruction

Finally, we applied our quartet-block pairs to reconstruct full tree topologies for the above sets of benchmark sequences. Here, we used two different approaches, namely *Quartet MaxCut* and *Maximum-Parsimony*, as described above. As is common practice in the field, we evaluated the quality of the reconstructed phylogenies by comparing the the respective reference trees from *AFproject* using the *normalized Robinson-Foulds (RF) distances* between the inferred and the reference topologies. For a data set with $N$ taxa, the *normalized RF distances* are obtained from the *RF distances* by dividing them by $2 * N - 6$, i.e. by the maximum possible *distance* for trees with $n$ leaves. The results of other alignment-free methods on these data are reported in [41, 47].

We applied the program *Quartet MaxCut* first to the quartet topologies derived from the set of *all* informative quartet-block pairs. As a comparison, we then inferred topologies using only those quartet-block pairs that *strongly* support one of the three possible topologies for the four involved sequences. The results of these test runs are shown in Table 3. Next, we used the program *PAUP** [6] to calculate the most parsimonious tree, using the distances between quartet blocks as characters, as explained above. Here, we used the *TBR* [56] heuristic. In some cases, this resulted in multiple optimal, i.e. most parsimonious trees. In these cases, we somewhat arbitrarily picked the first of these trees in the *PAUP** output. The results of these test runs are also shown in Table 3, together with the results from *Multi-SpaM*.

**Table 3. Average *normalized Robinson-Foulds (RF)* distances between trees, reconstructed with various alignment-free methods, for the genome sets listed in Table 2.** For each data sets, we performed 10 program runs with our indel-based approach. The average over these program runs is shown in the table; standard deviations are shown in parentheses.

| | *Gap-SpaM* | | | | *Multi-SpaM* | *FSWM* |
|---|---|---|---|---|---|---|
| | Quartet MaxCut | | | Parsimony | | |
| | **Strong** | **Weak** | **Combined** | | | |
| *E. coli* 29 | 0.21 (0.04) | 0.42 (0.06) | 0.22 (0.08) | 0.19 (0.04) | 0.24 | 0.12 |
| *E. coli* 27 | 0.21 (0.04) | 0.39 (0.07) | 0.19 (0.07) | 0.16 (0.06) | 0.18 | 0.17 |
| Fish mito | 0.44 (0.09) | 0.58 (0.1) | 0.41 (0.08) | 0.45 (0.07) | 0.18 | 0.05 |
| *Wolbachia* | 0.18 (0.04) | 0.25 (0.04) | 0.2 (0.03) | 0.18 (0.04) | 0.19 | 0.19 |
| *Yersinia* | 0.6 (0) | 0.9 (0.1) | 0.6 (0.07) | 0.6 (0) | 0.6 | 1 |
| Plants | 0.31 (0.07) | 0.35 (0.07) | 0.31 (0.08) | 0.3 (0.08) | 0.27 | 0.27 |
| *Piroplasmida* mito | 0.78 (0.1) | 0.88 (0.06) | 0.86 (0.09) | 0.84 (0.07) | 0.41 | 0.44 |
| *Termites* mito | 0.74 (0.21) | 0.74 (0.1) | 0.66 (0.08) | 0.73 (0.09) | 0.37 | 0.56 |

**Table 4. Program runtime for the approach described in this paper, in comparison to *Multi-SpaM*.** Column *reference blocks* contains the time to calculate the set of reference quartet blocks with the program *Multi-SpaM*. Column *gaps* contains the remaining runtime of our method, i.e. the time to find the respective second block for each reference block. As a comparison, column *RAxML* contains the runtime for running *RAxML* on the *don't-care* positions of the *reference blocks*.

| | reference blocks | RAxML | gaps |
|---|---|---|---|
| *E. coli* 29 | 119.17s | 428.42s | 86.57s |
| *E. coli* 27 | 114.36s | 437.85 | 87.12s |
| Fish mito | 0.86s | 19.15s | 2.67s |
| *Wolbachia* | 111.59s | 283.21s | 106.86s |
| *Yersinia* | 67.86s | 83.77s | 46.25 |
| Plants | 112,329.45s | 417.15s | 101.74s |
| *Piroplasmida* mito | 0.14s | 1.5s | 0.24s |
| *Termites* mito | 0.49s | 6.11s | 0.69s |

## 3.3 Runtime

The program runtime of our approach on the data sets that we used in our evaluation is shown in Table 4. Test runs were performed on an *Intel Xeon Processor E7- 4850* with 2.00 *GHz* (4 processors with 10 kernels each/20 threads) and *1 TB RAM*. Here, the runtime is shown separately for identifying the *reference blocks* by running the corresponding sub-routine of *Multi-SpaM* (first column) and for finding a second block for each reference block (third column). The time to calculate the resulting tree from the distances between these block pairs with *parsimony* or *Quartet MaxCut* was negligible. The total runtime of our approach is, thus, roughly the sum of the values in the first and the third column. As a comparison, we report the runtime of the well-known program *RAxML* [1] that *Multi-SpaM* uses on the don't-care positions of the reference blocks. Thus, the total run time of *Multi-Spam* is obtained as the sum of the values of the first two columns.

## 4 Discussion

Sequence-based phylogeny reconstruction usually relies on nucleotide or amino-acid residues aligned to each other in multiple alignments. Information about insertions and deletions (indels) is neglected in most studies, despite evidence that this information may be useful for phylogeny inference. There are several difficulties when indels are to be used as phylogenetic

signal: it is difficult to derive probabilistic models for insertions and deletions, and there are computational issues if gaps of different lengths are spanning multiple columns in multiple alignments. Finally, gapped regions in sequence alignments are often considered less reliable than un-gapped regions, so the precise number and length of insertions and deletions that have happened may not be easy to infer from multiple alignments.

In recent years, many fast alignment-free methods have been proposed to tackle the ever increasing amount of sequence data. Most of these methods are based on counting or comparing *words*, and gaps are usually not allowed within these words. It is therefore not straight-forward to adapt standard alignment-free methods to use indels as phylogenetic information.

In the present paper, we proposed to use *pairs of blocks* of sequence segments, based on our previously proposed alignment-free *spaced-word* approach. *Within* such blocks, no gaps are allowed. These blocks can be used to obtain information about possible insertions and deletions *between* two blocks since the compared sequences have evolved from a common ancestor. To this end, we consider the distances between these blocks in the respective sequences. If these distances are different for two sequences, this indicates that there has been an insertion or deletion since they evolved from their last common ancestor. If the two distances are the same, no indel event needs to be assumed. This information can be used to infer a tree topology for the sequences involved in a pair of blocks. To our knowledge, this is the first attempt to use insertions and deletions as phylogenetic signal in an alignment-free context.

In this study, we restricted ourselves, for simplicity, to *quartet blocks* i.e. to blocks involving *four* input sequences each; we used pairs of blocks involving the same four sequences. We did not consider the *length* of hypothetical insertions and deletions, but only asked whether or not such an event has to be assumed between two sequences in the region bounded by the two blocks. Since indels are relatively rare events, compared to substitutions, the *maximum parsimony* paradigm seems to be suitable in this situation. In the sense of *parsimony*, however, only those block pairs are informative that, in our notation, *strongly* support one of three possible quartet topologies, in the sense of the definition that we introduced in this paper. Indeed, if two distances between two blocks are equal, and the third and fourth distance are different from them–and also different from each other –, then each of the three possible quartet topologies would require two insertion or deletion events. That is, all three topologies would be equally good from a parsimonious viewpoint.

Intuitively, however, one may want to use the information from such quartet blocks pairs that, in our terminology, *weakly* support one of the possible topologies. It is easy to see that, with a simple probabilistic model under which an insertion between two blocks occurs with a probability $p < 0.5$, independently of the length of the insertion and the distance between the blocks, a *weakly* supported topology would have a higher likelihood than the two alternative topologies—although all three topologies are considered equally good from a parsimony point-of-view. So it might be interesting to apply such a simple probabilistic model to our approach, instead of maximum parsimony. Also, while we restricted ourselves to quartet blocks in this study, it might be worthwhile to use block pairs involving more than four sequences.

Our approach has only few parameters that can be adjusted by the user, essentially concerning the underlying binary pattern $P$ and the threshold that is used to separate random quartet blocks form quartet blocks that represent true homologies. In our study, we used patterns with a length of 110 and with 10 match positions, i.e. with 100 don't-care positions. Our results in the present and in previous studies indicate that with our default parameter value, random spaced-word matches can be reliably distinguished from background matches [34, 41]. Adapting these parameter values mainly affects the *number* of quartet-block pairs. So this mainly

comes down to a trade off between program run time and the amount of information that one obtains form the block pairs, i.e. the strength of the signal.

Using standard benchmark data, we could show that phylogenetic signal from putative insertions and deletions between quartet blocks is mostly in accordance with the reference phylogenies that we used as standard of truth. Interestingly, the quality of the tree topologies that we constructed from our 'informative' pairs of quartet blocks—i.e. from indel information alone—is roughly comparable to the quality of topologies obtained with existing alignment-free methods.

As mentioned above, our approach is not competing with existing phylogeny approaches. In fact, we did not expect to obtain trees with our approach that are of comparable quality as trees obtained with standard methods. Our goal was to find out if information about putative insertions and deletions can provide useful phylogenetic information at all in an alignment-free setting. Since the phylogenetic signal from indels is complementary to the information that is used by those existing approaches, any such information might be useful as additional evidence, no matter if substitution-based or indel-based trees are superior. It is all the more surprising that our rather simplistic approach is already able to infer trees that are roughly comparable to trees obtained with established alignment-free approaches.

There is a certain limitation of our approach, if it is used as a *stand-alone approach*, to infer trees without additional information, as we did in our evaluation: To infer trees from putative indels alone, we need a large enough number of 'informative' block pairs to obtain *quartet trees*, or as an input *for maximum parsimony*. The number of informative block pairs that we can obtain depends, however, on the sequence length and on the degree of similarity among the compared sequences. If sequences are to short or to distantly related, our approach cannot find sufficiently many *reference quartet blocks* with a score above the employed threshold value.

In our test runs, we used three data sets with a low degree of similarity, the *plants* data set and the mitochondrial genomes from *Piroplasmida* and *Termites*, see Table 1. The plant genomes that we used were large enough to obtain a sufficient number of informative block pairs and, as a result, a phylogenetic tree that is of comparable quality as the tree produced with *FSWM* and *Multi-SpaM*. The trees we obtained from the *Piroplasmida* and *Termites* mitochondrial DNA were of poor quality, though. Note however, that even for these two data sets, a rather large fraction of the 'strongly informative' block pairs were in accordance with the reference phylogeny, namely 63.78 and 47.98 percent, respectively, see Table 2. This indicates that, while these block pairs are not sufficient to infer the correct tree topology, when used as the sole source of information, they may still be useful as *additional* input information when combined with other approaches to phylogeny reconstruction. Therefore, it seems worthwhile to investigate how our indel-based approach can be used together with other alignment-free approaches.

## Author Contributions

**Conceptualization:** Burkhard Morgenstern.

**Formal analysis:** Burkhard Morgenstern.

**Funding acquisition:** Burkhard Morgenstern.

**Investigation:** Niklas Birth, Burkhard Morgenstern.

**Methodology:** Niklas Birth, Thomas Dencker, Burkhard Morgenstern.

**Resources:** Thomas Dencker.

**Software:** Niklas Birth, Thomas Dencker.

**Supervision:** Burkhard Morgenstern.

**Validation:** Niklas Birth.

**Writing – original draft:** Thomas Dencker, Burkhard Morgenstern.

**Writing – review & editing:** Niklas Birth, Burkhard Morgenstern.

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
