## [Decision Letter · Decision Letter 0]

31 Jan 2022

Dear Prof. Morgenstern,

Thank you very much for submitting your manuscript "Insertions and deletions as phylogenetic signal in alignment-free sequence comparison" for consideration at PLOS Computational Biology.

As with all papers reviewed by the journal, your manuscript was reviewed by members of the editorial board and by several independent reviewers. In light of the reviews (below this email), we would like to invite the resubmission of a significantly-revised version that takes into account the reviewers' and editor's comments.

The assumption that “We assume that the blocks B1 and B2 are representing true homologies,” needs much clarification and some examples to support. The last common ancestor may be identified as a consequence of indels in the sequences. B1 and B2 themselves contains many positions which are not considered in the process. All of these concerns need clarifications. The method to identify B1 and B2 may be heavily affected by the imbalance of the data. In addition, the quartet approach is good for homologous sequences, but whether it is good for heterogeneous sequence is under-investigated. More experiments are needed to illustrate these points.

We cannot make any decision about publication until we have seen the revised manuscript and your response to the reviewers' comments. Your revised manuscript is also likely to be sent to reviewers for further evaluation.

Sincerely,

Jinyan Li

Associate Editor

PLOS Computational Biology

Thomas Leitner

Deputy Editor

PLOS Computational Biology

The assumption that “We assume that the blocks B1 and B2 are representing true homologies,” needs much clarification and some examples to support. The last common ancestor may be identified as a consequence of indels in the sequences. B1 and B2 themselves contains many positions which are not considered in the process. All of these concerns need clarifications. The method to identify B1 and B2 may be heavily affected by the imbalance of the data. In addition, the quartet approach is good for homologous sequences, but whether it is good for heterogeneous sequence is under-investigated. More experiments are needed to illustrate these points.

Reviewer's Responses to Questions

**Comments to the Authors:**

Reviewer #1: Birt et al. propose a method for phylogeny reconstruction based on the

phylogenies supported by a large number of quartets of sequences. They

calculate the quartet phylogenies from indels between quartet blocks

identified through pattern matches. If the distances between the

patterns in a pair of blocks differ, an indel is inferred. The authors

apply the new method to six samples of genomes and summarize their

results in Table 2, where their new method returns trees that may

compete with FSWM, another pattern-based method by the Morgenstern

group.

It is well-known that indels carry phylogenetic signal, but to measure

them without standard alignment is an interesting new direction. The

paper might be further improved by considering the following three

comments:

1) In Table 2, combining the weak and the strong quartet pairs seems

to reduce the overall quality of the result. Perhaps the method should

be restricted to the strong quartets?

2) Table 2 lists averages. What was the variability around those

averages?

3) The paper is making a contricution to alignment-free sequence

comparison, where minimizing resource consumption compared to

alignment has traditionally been a central concern. However, I

couldn't find any measurements on the resource consumption of the new

method.

**Have the authors made all data and (if applicable) computational code underlying the findings in their manuscript fully available?**

Reviewer #1: Yes

PLOS authors have the option to publish the peer review history of their article (what does this mean?). If published, this will include your full peer review and any attached files.

Reviewer #1: No
---

## [Decision Letter · Decision Letter 1]

14 Jun 2022

Dear Prof. Morgenstern,

We are pleased to inform you that your manuscript 'Insertions and deletions as phylogenetic signal in alignment-free sequence comparison' has been provisionally accepted for publication in PLOS Computational Biology.

Best regards,

Jinyan Li

Associate Editor

PLOS Computational Biology

Thomas Leitner

Deputy Editor

PLOS Computational Biology

The reviewer was satisfied with the revised version of the manuscript; and the editor is also satisfied with the author's reply and the revised manuscript. However, there are obvious typos in the manuscript, for example "to short" is a typo. Other typos include "to distantly", "the the". These typos must be corrected during the paper proof-reading stage.

Reviewer's Responses to Questions

**Comments to the Authors:**

Reviewer #1: The authors have addressed my comments, I am happy with their replies and have no further comments.

**Have the authors made all data and (if applicable) computational code underlying the findings in their manuscript fully available?**

Reviewer #1: Yes

PLOS authors have the option to publish the peer review history of their article (what does this mean?). If published, this will include your full peer review and any attached files.

Reviewer #1: No

---

## [Editor Report · Acceptance letter]

26 Jul 2022

PCOMPBIOL-D-21-01893R1 

Insertions and deletions as phylogenetic signal in an alignment-free context

Dear Dr Morgenstern,

I am pleased to inform you that your manuscript has been formally accepted for publication in PLOS Computational Biology. Your manuscript is now with our production department and you will be notified of the publication date in due course.

With kind regards,

Zsuzsanna Gémesi
